# Halide Perovskite Inducing Anomalous Nonvolatile Polarization in Poly(vinylidene fluoride)-based Flexible Nanocomposites

Yao Wang [1] ✉, Chen Huang[1], Ziwei Cheng[1], Zhenghao Liu[2,3], Yuan Zhang[2,3], Yantao Zheng[1], Shulin Chen[4], Jie Wang[5], Peng Gao[4,6], Yang Shen[7], Chungang Duan[8], Yuan Deng [9], Ce-Wen Nan [7] & Jiangyu Li [2,3] ✉

Ferroelectric materials have important applications in transduction, data storage, and nonlinear optics. Inorganic ferroelectrics such as lead zirconate titanate possess large polarization, though they are rigid and brittle. Ferroelectric polymers are light weight and flexible, yet their polarization is low, bottlenecked at 10 $\mu C\ cm^{-2}$. Here we show poly(vinylidene fluoride) nanocomposite with only 0.94% of self-nucleated $CH_3NH_3PbBr_3$ nanocrystals exhibits anomalously large polarization (~19.6 $\mu C\ cm^{-2}$) while retaining superior stretchability and photoluminance, resulting in unprecedented electromechanical figures of merit among ferroelectrics. Comprehensive analysis suggests the enhancement is accomplished via delicate defect engineering, with field-induced Frenkel pairs in halide perovskite stabilized by the poled ferroelectric polymer through interfacial coupling. The strategy is general, working in poly(vinylidene fluoride-co-hexafluoropropylene) as well, and the nanocomposite is stable. The study thus presents a solution for overcoming the electromechanical dilemma of ferroelectrics while enabling additional optic-activity, ideal for multifunctional flexible electronics applications.

Ferroelectrics are crystalline materials with spontaneous polarization that can be reversed by an external electric field, making them indispensable for a wide range of applications in electromechanical transduction, data storage, nonlinear optics, and solid-state cooling[1–5]. Inorganic oxides such as lead zirconate titanate (PZT) possess large polarization and strong piezoelectricity, though their rigidity and brittleness severely hinder their applications in emerging flexible electronics[6]. For example, Young's modulus of PZT ceramics is rather high, in the order of 100 GPa, and its fracture strain is quite small, no more than 0.2%[7]. The discovery of ferroelectric poly(vinylidene fluoride) (PVDF) in 1969[8] and subsequently its copolymer with trifluoroethylene (P(VDF-TrFE))[9] has greatly expanded the applications of ferroelectrics, thanks to their flexibility, lightweight, toughness, and easy processibility[10]. In the last 50 years, substantial advances have

[1]School of Materials Science and Engineering, Beihang University, Beijing 100191, China. [2]Department of Materials Science and Engineering, Southern University of Science and Technology, Shenzhen 518055 Guangdong, China. [3]Guangdong Provincial Key Laboratory of Functional Oxide Materials and Devices, Southern University of Science and Technology, Shenzhen 518055 Guangdong, China. [4]International Center for Quantum Materials and Electron Microscopy Laboratory, School of Physics, Peking University, Beijing 100871, China. [5]Department of Engineering Mechanics, Key Laboratory of Soft Machines and Smart Devices of Zhejiang Province, Zhejiang University, Hangzhou 310027 Zhejiang, China. [6]Collaborative Innovation Center of Quantum Matter, Beijing 100871, China. [7]School of Materials Science and Engineering, State Key Lab of New Ceramics and Fine Processing, Tsinghua University, Beijing 100084, China. [8]State Key Laboratory of Precision Spectroscopy and Key Laboratory of Polar Materials and Devices, Ministry of Education, Department of Electronics, East China Normal University, Shanghai 200241, China. [9]Key Laboratory of Intelligent Sensing Materials and Chip Integration Technology of Zhejiang Province, Hangzhou Innovation Institute, Beihang University, Hangzhou 310052 Zhejiang, China. ✉e-mail: wang-yao@buaa.edu.cn; lijy@sustech.edu.cn

been achieved on their electromechanical coupling via strategies such as forming relaxors[11], nevertheless, there is no significant improvement in the polarization of polymeric ferroelectrics, bottlenecked at less than 10 μC cm$^{-2}$[12,13]. No effective strategy has been put forward to improve its polarization significantly, and despite extensive efforts there is only limited success achieved, for example, through morphotropic phase boundary[14] or biaxial orientation[15]. Indeed, the large polarization in a ferroelectric is often accompanied by high stiffness, making soft ferroelectrics with strong polarization elusive. In recent years, molecular ferroelectrics have reemerged as a promising alternative to ferroelectric polymers[16], though the competing demands on both strong polarization and large mechanical compliance have yet to be satisfied.

In this work, we overcome this dilemma by incorporating CH$_3$NH$_3$PbBr$_3$ (MAPB) nanocrystals into the PVDF matrix. While a composite approach incorporating a large volume fraction of ferroelectric oxide fillers has been widely used to enhance the electromechanical properties of polymers[17], it also increases stiffness substantially while decreasing stretchability and breakdown strength[18], defeating the very purpose of polymeric ferroelectrics. We solve this problem by using less than 1% volume fraction of MPAB self-nucleated in the PVDF matrix. At the first sight, choosing organic-inorganic hybrid perovskites MAPbX$_3$ (MAPX; MA = methylammonium, X = Cl, Br, or I) as the second phase filler for PVDF seems quite counterintuitive due to its small if any polarization[19–23]. Furthermore, the unstable nature of MAPX also raises serious challenges[24,25]. A more deliberate contemplation, however, suggests the ionic migration that is responsible for their instability[26,27] also induces defect dipoles in materials[28], and thus it may lead to dipolar order under appropriate tuning given its excellent polarizability. The key thus lies in making such defect dipoles nonvolatile, which we believe can be stabilized by the PVDF matrix, and this is essentially our strategy. Meanwhile, ferroelectric materials are also responsive to light, presenting electro-optic, photovoltaic, and other photoexcitation-related physical processes, which have drawn extensive attention showing significant application potentials[29,30]. Introducing MAPB endows the ferroelectric polymer with optic-active functions, which have rarely been addressed in ferroelectric polymers.

Indeed, compositing MAPB results in enhanced polarization of PVDF almost threefold while retaining its intrinsic flexibility and stretchability, exhibiting strong photoluminescence as well. Quite remarkably, these nanocomposites are highly stable, despite using intrinsically unstable MAPB[25], and the strategy turns out to be very effective and quite general, working equally well for poly(vinylidene fluoride-co-hexafluoropropylene) [P(VDF-HFP)] with polar bonding[31]. The work thus not only provides an innovative solution for polymeric ferroelectrics with substantially enhanced properties and multifunctionality but also points toward a possible direction to improve the stability of MAPX.

## Results

### Microstructure and optic-activity

High-quality MAPB/PVDF nanocomposite films were synthesized via a growth-confined in situ nucleation process as illustrated in Fig. 1a. The key to our synthesis is that nucleation of MAPB nanocrystals and formation of PVDF polymer matrix is almost simultaneous, as revealed by Supplementary Movie 1 and Supplementary Fig. 1a, b, wherein crystallization of PVDF starts before 20 min as indicated by gradual loss of transparency of as-cast film, while nucleation of MAPB nanocrystals completes at about 20 min as indicated by the stable photoluminescence intensity. This ensures good dispersion of MAPB nanocrystals inside PVDF since the free volume of PVDF polymer provides the space for MAPB nucleation while the simultaneous film formation confines further crystal growth as well as agglomeration. It also results in cohesive binding between MAPB and PVDF matrix, overcoming major difficulties associated with the processing of PVDF-based nanocomposites[32], especially those involving unstable materials such as halide perovskites, thanks to the strong molecular interaction enabled by highly polar C–F bond with strong electronegativity and unique hybrid ABX$_3$ molecular structure containing methylamine cations. As confirmed by the low magnification transmission electron microscopy (TEM, Fig. 1b), the microstructure of nanocomposite film

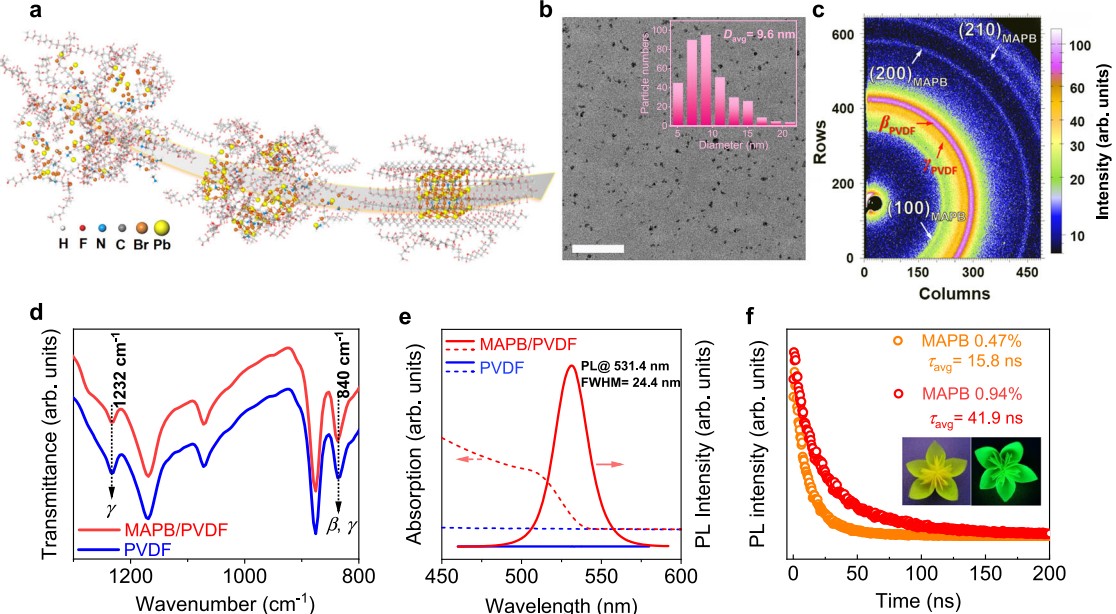

**Fig. 1 | Structures and optical properties of MAPB/PVDF nanocomposite in comparison with PVDF. a** Schematic illustration of the confined growth of MAPB in a PVDF matrix. **b** TEM image of nanocomposite film having 0.94% MAPB nanocrystals with the inset showing the size distribution of MAPB nanocrystals and the average particle diameter ($D_{avg}$). The scale bar is 200 nm. **c** 2D WAXS pattern of nanocomposite film with the diffraction rings from β-phase PVDF ($β_{PVDF}$), γ-phase PVDF ($γ_{PVDF}$) and MAPB nanocrystals assigned, respectively. **d** FTIR spectra, **e** UV–vis absorption and photoluminescence spectra, and (**f**) PL decay spectra, where the insets show the photos of flower-shaped films illuminated under sunlight and ultraviolet, respectively.

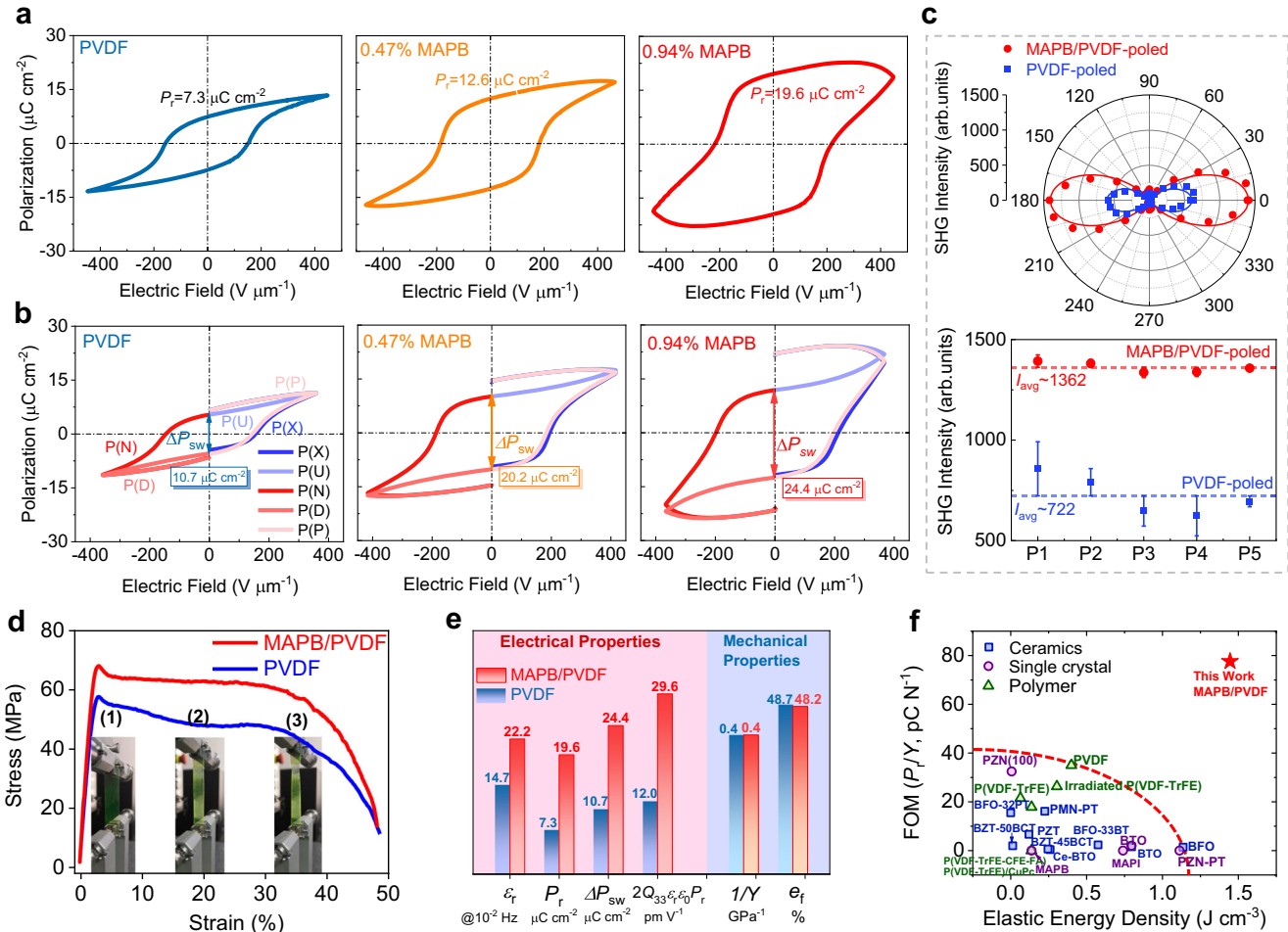

**Fig. 2 | Ferroelectric polarization of MAPB/PVDF nanocomposites in comparison to PVDF film.** **a** Polarization-electric field hysteresis loop measured at 10 Hz. **b** PUND measurements at 10 Hz. **c** Polar plots of transmitted SHG intensity for poled films with the solid lines fitting based on an orthogonal *mm2* polar structure and multiple points statistics on SHG intensities. The average values ($I_{avg}$) are obtained from multiple points tests at five different regions of the films and the error bars are the deviations from the average. **d** Stress-strain curves with the insets showing the photos of nanocomposite film under different strains of 0%, 33%, and 41%, respectively. **e** Comparison of electromechanical properties between nanocomposite film and PVDF polymer: permittivity @ 0.01 Hz, $P_r$, $\Delta P_{sw}$ obtained from PUND measurement, calculated $d_{33}$ values based on electrostriction ($2Q_{33}\varepsilon_r\varepsilon_0 P_r$), mechanical compliance (1/Y), and fracture strain ($e_f$). **f** Comparison of the figure of merits (FOM: $P_r/Y$) versus electromechanical energy density for different materials reported in literature[11,28,38–48] and this study. The dash line is used to guide the eye.

shows monodisperse MAPB nanocrystals in PVDF matrix with an average particle diameter of 9.6 nm, which is further supported by higher magnification TEM as well as high-resolution TEM (HRTEM) images in Supplementary Fig. 2. The nanocomposite film appears semi-transparent with light yellow–green color as shown by Supplementary Fig. 1c, and note that homogenous nanocomposite film as large as 150 mm × 120 mm was formed in one step (Supplementary Movie 1). The crystalline structure of each phase in the nanocomposite is identified via wide-angle X-ray scattering (WAXS, Fig. 1c), revealing collectively polycrystalline perovskite phase of MAPB and polar $\beta$ and $\gamma$ phases of PVDF. The crystallinities of pure PVDF and nanocomposite films were estimated based on differential scanning calorimetry (DSC) curves (Supplementary Fig. 3), and the addition of a small fraction of MAPB nanocrystals enhances the crystallinity of PVDF slightly from 45.7% to 51.2%. Fourier transform infrared spectra (FTIR) (Fig. 1d) indicate that there is no detectable phase transition involved in PVDF induced by MAPB, consistent with the phase analysis via X-ray and Raman spectra (Supplementary Fig. 4).

It is worth noting that the nanocomposite film exhibits good light-emitting properties (Fig. 1e, f) arising from MAPB nanocrystals even at ultralow MAPB volume fraction, i.e., 0.47%, which is also demonstrated by the green fluorescence photo under ultraviolet illumination in the

inset of Fig. 1f. The absorption onset of MAPB/PVDF nanocomposite occurs at around 540 nm, while the photoluminescence (PL) emission peak occurs at around 530 nm with full width at half maximum (FWHM) 24.4 nm, consistent with the band-to-band transition of bromide perovskite[33] and those measured from colloidal MAPB nanoparticles (Supplementary Fig. 5). Time-resolved PL decay spectra are shown in Fig. 1f, with calculated average lifetime ($\tau_{avg}$) 15.8 ns for 0.47% MAPB/PVDF and 41.9 ns for 0.94% MAPB/PVDF nanocomposites, respectively. This set of data thus unambiguously established growth-confined monodispersed MAPB nanocrystals in a PVDF matrix with cohesive interfacial bonding, rendering MAPB/PVDF a photoactive functionality that is absent in pure PVDF. Such photoactivity may be used for anti-counterfeiting ferroelectric memory, and may even enable novel multi-bit data storage. Additional applications, as well as new phenomena, may also emerge from the interplay between polarization and light, which is currently under investigation.

### Ferroelectricity and electromechanical performance

Despite that only less than 1% of MAPB nanocrystals are incorporated in PVDF matrix, we observe anomalously enhanced ferroelectric behaviors as summarized in Fig. 2. The polarization hysteresis loops of MAPB/PVDF in comparison to pure PVDF film (Fig. 2a) reveal that the

remnant polarization ($P_r$) increases from 7.3 for pure PVDF film to 19.6 μC cm$^{-2}$ for 0.94% MAPB/PVDF nanocomposites, respectively, and this large polarization is nearly threefold the value of typical ferro-electric PVDF film, far exceeding that of ferroelectric copolymer P(VDF-TrFE) with typical value less than 10 μC cm$^{-2}$[13,34]. Remarkably, the ferroelectricity of MAPB/PVDF is highly stable, with the polarization-electric field ($P$–$E$) loop almost unchanged when measured after 50 days (Supplementary Fig. 6), retaining ~90% of initial $P_r$ with almost identical coercive voltage. The photoluminescence property of the nanocomposite is also quite stable and hardly changes after 10 days in water (Supplementary Fig. 7). Such excellent stability is quite unusual for halide perovskites. The key may lie in the well-monodispersed MAPB nanocrystals in the PVDF matrix, whose strong C−F bond has a crucial impact on the high resistance to oxidation and hydrolytic stability[12], thus effectively protecting MAPB nanocrystals from decomposition.

To elaborate the mechanism responsible for the apparent polarization measured for MAPB/PVDF nanocomposites, Positive-Up Negative-Down (PUND) test[35] has been carried out (Fig. 2b), elucidating intrinsic switched polarization ($\Delta P_{sw}$) of 20.2 and 24.4 μC cm$^{-2}$ for MAPB/PVDF nanocomposite with 0.47% and 0.94% of MAPB, respectively, while that of pure PVDF film is only 10.7 μC cm$^{-2}$. As a result, such anomalously enhanced polarization is intrinsic. Correspondingly, we also observe a large increase in permittivity at low frequencies (Supplementary Fig. 8), increasing from 14.7 for pure PVDF to 22.2 for MAPB/PVDF nanocomposite measured at 0.01 Hz, while they converge at higher frequencies beyond 10 Hz. Due to the heterogeneous nature of semi-crystalline PVDF, the motion of charges necessary for compensating the polarization in the crystallite phase leads to a low-frequency dispersion, and the enhanced permittivity is consistent with the large polarization.

To further confirm that the large polarization indeed arises from long-range polar order breaking the crystalline symmetry, optical second harmonic generation (SHG) technique has been employed[36] (Supplementary Fig. 9a). Transmitted SHG signals generated from both poled MAPB/PVDF and PVDF films with respect to incident light polarization angle are presented in Fig. 2c, in sharp contrast to the as-prepared films showing negligible SHG responses (Supplementary Fig. 9b). The $p$-out SHG polar plots of both films display a twofold symmetry, which is well fitted by an orthogonal $mm2$ structure (Supplementary Note 7). Notice that the average SHG intensity of poled MAPB/PVDF is twice the value of poled PVDF (Fig. 2c, Supplementary Fig. 9c, d), indicating substantially enhanced remnant polarization $P_r$ in MAPB/PVDF nanocomposite, consistent with the $P$–$E$ and PUND results.

Unlike the generally employed inorganic ferroelectric ceramic fillers, hybrid halide perovskites have an order of magnitude smaller elastic modulus[37], and thus the addition of MAPB nanocrystals favors maintaining the macroscopic mechanical property of PVDF. As presented in Fig. 2d, the stress-strain curve of nanocomposite retains the good stretchability and flexibility of PVDF film, with the initial linear region revealing Young's modulus of 2.5 GPa followed by nonlinear deformation in both polymer and nanocomposite films. The fracture strain is as large as 48%, which is two orders of magnitude higher than PZT. In other words, the polarization and electromechanical coupling of nanocomposites are enhanced substantially without compromising the superior flexibility and stretchability of PVDF polymer as summarized in Fig. 2e, wherein $P_r$ and $\Delta P_{sw}$ both exhibit almost 250% increase, while the piezoelectric coefficient $d_{33}$ estimated based on electrostriction $2Q_{33}\varepsilon_r\varepsilon_0 P_r$ shows comparable enhancement, increasing from 12.0 pm V$^{-1}$ for pure PVDF to 29. 6 pm V$^{-1}$ for MAPB/PVDF nanocomposite, with $Q_{33}$ being the electrostrictive coefficient estimated to be −1.127 m$^4$ C$^{-2}$ (Supplementary Fig. 10). Meanwhile, both Young's modulus ($Y$) and fracture strain ($e_f$) remain essentially

unchanged. For flexible ferroelectric applications, $P_r/Y$ is one of the most important figures of merit, measuring the polarization retention against the capability of mechanical deformation, and the MAPB/PVDF nanocomposite film exhibits the champion $P_r/Y$ value of 77.6 pC N$^{-1}$ among reported ferroelectrics, as shown in Fig. 2f, outperforming known ferroelectric crystals, ceramics, as well as polymers[11,28,38–48]. Despite this flexibility, it is also capable of delivering an impressive electromechanical energy density $YS_m^2/2$ of 1.445 J cm$^{-3}$, surpassing other ferroelectric materials as well and comparable to that of muscle[49]. Such large polarization is also quite stable under stretch. The $P$–$E$ loop is insensitive to deformation as high as 5% (Supplementary Fig. 11).

The increases in polarization and electromechanical coupling presented in Fig. 2 are not only substantial but also anomalous and puzzling. With only 0.94% of MAPB, the enhancement cannot be explained by the existing mean-field or effective medium theories[50]. There were previous studies reporting slight improvement in the polarization of PVDF-based nanocomposites, which was generally ascribed to the increase of polar $\beta$ phase[51–53], though we see no obvious nonpolar-polar phase transition observed as evidenced from WAXS, FTIR and Raman spectra shown in Fig. 1 and Supplementary Fig. 4. The slight increase in PVDF crystallinity (Supplementary Fig. 3) could not explain the large enhancement in polarization either. We thus turn to microscopic piezoresponse force microscopy (PFM) studies for further insight. We first mapped PFM response for PVDF and MAPB/PVDF films before and after poling, as shown in Fig. 3a–d, and observed only minor increase in piezoresponse of MAPB/PVDF (Fig. 3c) in comparison to PVDF (Fig. 3a) before poling both samples. Thing changes dramatically, however, after poling, where substantial enhancement is observed in MAPB/PVDF (Fig. 3d) in comparison to PVDF (Fig. 3b), consistent with the macroscopic measurement in Fig. 2. In order to understand the difference made by poling on MAPB/PVDF, we compare its first and second harmonic piezoresponses[54] measured before and after poling, as summarized by the histogram distributions in Fig. 3e, f obtained from the corresponding spatial mappings (Fig. 3c, d and Supplementary Fig. 12). Quite remarkably, it is observed that second harmonic response in MAPB/PVDF is much larger than first harmonic one before poling (Fig. 3e), yet it is greatly reduced after poling. Meanwhile, the first harmonic response is enhanced substantially after poling, becoming clearly dominating the second harmonic one (Fig. 3f). The enhanced linear response is also made more evident by comparing the first harmonic piezoresponse of PVDF and MAPB/PVDF versus AC voltage before and after poling, revealing much-increased slope, and thus the effective $d_{33}$, in MAPB/PVDF after poling (Fig. 3g), while not much difference is seen between MAPB/PVDF before poling and pure PVDF w/o poling. This set of data suggests that a large poling field is indispensable to induce large polarization and electromechanical coupling for the nanocomposite. In particular, the small AC voltage of PFM measurement induces a dipole moment in MAPB/PVDF that is volatile, making its electromechanical response predominantly nonlinear quadratic, while the large DC poling makes the induced dipole moment nonvolatile, resulting in a predominantly linear electromechanical response biased by the large dipole moment. This also explains the crossover between the first and second harmonic piezoresponse before and after poling in Fig. 3e, f. The comparisons of piezoresponse and effective $d_{33}$ among poled and unpoled PVDF and MAPB/PVDF are summarized in Fig. 3h, and the poled MAPB/PVDF clearly stands out, with the piezoresponse increasing from 9.2 to 26.3 arb. units by poling, while the effective $d_{33}$ increases from 1.3 to 6.8 arb. units V$^{-1}$, both several times higher than 11.4 arb. units and 1.6 arb. units V$^{-1}$ of poled PVDF. This set of microscopic studies thus confirms the macroscopic data while revealing the critical role of poling for enhancement.

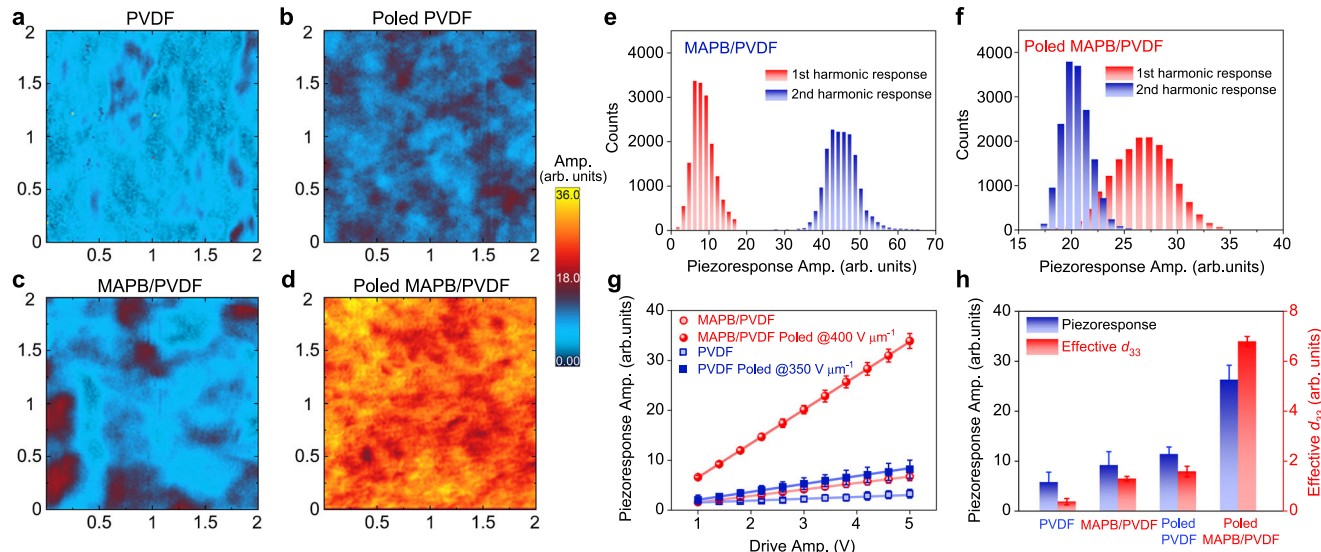

**Fig. 3 | Piezoelectric responses of MAPB/PVDF nanocomposite in comparison to PVDF film.** Piezoresponse amplitude mappings of (**a**) PVDF, **b** poled PVDF, **c** MAPB/PVDF, and **d** poled MAPB/PVDF. The scanning area is 2 μm × 2 μm. Histogram distribution of first and second harmonic piezoresponse amplitude for MAPB/PVDF nanocomposite (**e**) before and **f** after poling. **g** AC voltages dependent variation of piezoresponse amplitude of MAPB/PVDF nanocomposite and PVDF before and after poling. The error bars are the standard deviations. **h** Comparison of piezoresponses measured under 6 V AC, and the effective $d_{33}$ between MAPB/PVDF and PVDF before and after poling. The error bars are the standard deviations.

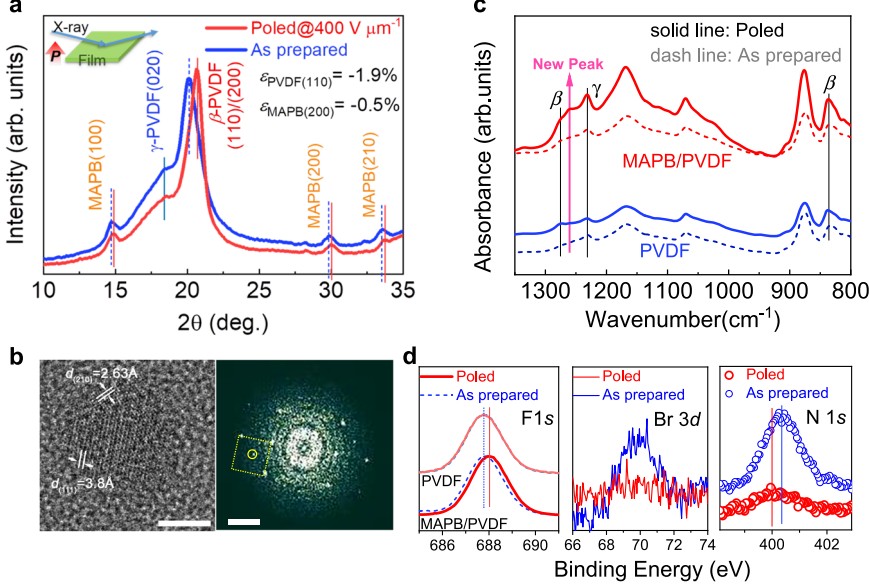

**Fig. 4 | Structural and chemical bonding variation in MAPB/PVDF nanocomposite induced by large poling field in comparison with PVDF film. a** WAXS measured at reflected mode with respect to the polarization direction indicated in the inset. **b** HRTEM image of poled MAPB/PVDF nanocomposite and corresponding FFT diffraction pattern. The scale bar is 5 nm on the left and 2 nm−1 on the right image, respectively. **c** FTIR spectra showing new vibrating mode emerging in MAPM/PVDF nanocomposite after poling, which is not observed in poled PVDF. **d** Comparison on XPS spectra of F 1$s$ in both PVDF and MAPB/PVDF, N 1$s$ and Br 3$d$ of MAPB/PVDF before and after poling.

## Microscopic mechanism

Since the poling-induced enhancement is much less in pure PVDF (Fig. 3h), while poled MAPB/PVDF exhibits a large dipole moment absent before poling, we hypothesize that the mechanism is related to Frenkel pairs in MAPB induced by an electric field, which has recently been reported in halide perovskites[28]. To verify this hypothesis, we investigated the effects of poling on the crystal lattices, phases, chemical bonds, and defects of MAPB and PVDF in detail. 1D WAXS patterns of poled MAPB/PVDF nanocomposite film in comparison with as-prepared film presented in Fig. 4a show evident shift to higher angle for both PVDF and MAPB lattices after poling, generating a large compressive strain of −1.9% due to its negative longitudinal piezoelectric coefficient[48] for the poled $\beta$-phase PVDF and a compression of −0.5% for MAPB nanocrystals. The electrostrictive behavior of MAPB nanocrystals within the PVDF matrix is consistent with the observed large electrostriction in lead halide perovskites single crystals, where Frenkel dipoles were generated and stabilized by an electric field, accompanied by a lattice contraction[28]. High-resolution transmission electron microscopy (HRTEM) image of the poled MAPB/PVDF nanocomposite is shown in Fig. 4b along with the corresponding FFT

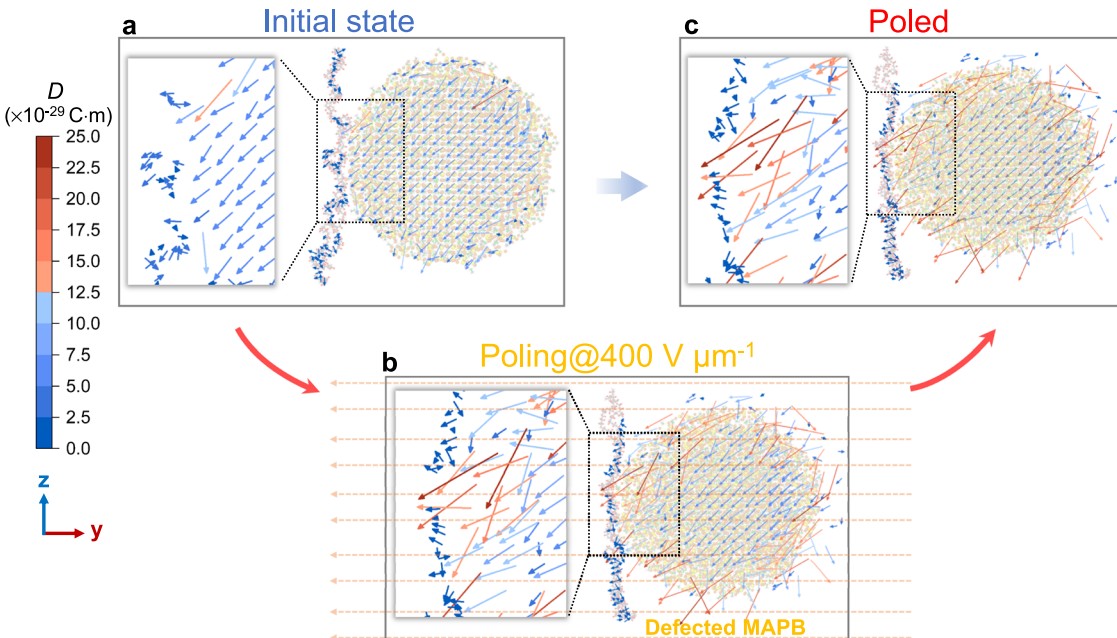

**Fig. 5 | Molecular dynamic simulations on the dipole moments distribution in MAPB/PVDF nanocomposites. a** Dipoles distribution of as-prepared MAPB/PVDF nanocomposite. **b** Dipoles distribution of MAPB/PVDF nanocomposites when $Br$ Frenkel defects are created in the MAPB nanocrystal during poling at $E = 400$ V μm$^{-1}$, and significantly strengthened polarization appears at the surface of MAPB nanocrystal with obvious interfacial coupling observed. **c** Dipoles distribution of MAPB/PVDF nanocomposite after poling electric field is withdrawn, and the dipoles from defected MAPB are retained, indicating poled PVDF provides an essential built-in electric field in stabilizing the defect structure in MAPB via interfacial coupling. The simulations show a slice from 3D dipole distribution in the yz plane at the same region, with the enlarged images displaying the dipole distribution around the interface between MAPB and PVDF.

diffraction pattern of a single MAPB nanocrystal, where superstructure spots have been observed as circled inside the normal MAPB diffraction spots. The emergence of such superstructure corresponds to the MAPB structure with Frenkel pair of $V^{\bullet}_{Br} - I'_{Br}$, formed between interstitial Br atom and Br vacancy as demonstrated from our previous studies[27].

Phase structure variation analyses via combined WAXS, FTIR and Raman spectra show that poling enhances polarized $\beta$-phase more significantly in PVDF than that of MAPB/PVDF film (Fig. 4a, c and Supplementary Figs. 13 and 14), which further support that polarized $\beta$-phase cannot accounts for the observed polarization enhancement. FTIR spectra reveal that a new vibrating mode has emerged in MAPB/PVDF nanocomposites after poling (Fig. 4c), which is absent in both poled PVDF and unpoled MAPB/PVDF nanocomposite, providing direct evidence for a strong interfacial coupling strengthened by poling-induced Frenkel pairs. This is further supported by atomic force microscopy-infrared spectroscopy (AFM-IR) mappings (Supplementary Fig. 15) showing strong IR responses to 1260 cm$^{-1}$ irradiation in poled MAPB/PVDF, yet no response in either poled PVDF or unpoled MAPB/PVDF. Raman spectra reveal that MAPB/PVDF nanocomposite retains $\gamma$-phase after poling (Supplementary Fig. 14b, c) in contrast to PVDF wherein $\gamma$-phase is partially transformed by poling (Supplementary Fig. 14a, c), further demonstrating the strong interfacial coupling between PVDF and MAPB. The shift of F 1s X-ray photoelectron spectra (XPS) to higher binding energy (Fig. 4d) in the MAPB/PVDF nanocomposite is observed, which is absent in PVDF after poling, suggesting electron transfer from fluorine atom to MAPB-induced by high voltage poling. The sharp decline in intensities for Br 3d and N 1s XPS signals after poling together with the slightly reduced optical bandgap of MAPB and shifting in PL peak to higher wavelength (Supplementary Fig. 16) also support the formation of Br vacancies in poled MAPB, generating extra positive charges and thus stronger bonding with F atoms in PVDF, favoring an ordered arrangement of $V^{\bullet}_{Br} - I'_{Br}$ dipoles toward poling direction.

It is quite possible that such ordered dipoles of $V^{\bullet}_{Br} - I'_{Br}$ can be arranged in ferroelectric ordering so as to induce giant polarization, and our molecular dynamics (MD) simulations indeed demonstrate how it works (Fig. 5, Supplementary Figs. 17–19). Compared to the initial state of as-prepared nanocomposite (Fig. 5a), when poled under a large electric field, dipole moments are aligned along the electric field direction, with their magnitude increasing from $5 \times 10^{-29}$ to $25 \times 10^{-29}$ C m on the surface of MAPB nanocrystal as Frenkel defects are created. In particular, a bromine ion within the Pb–Br octahedral frame migrates to one of its four adjacent tetrahedral interstices (Fig. 5b, Supplementary Fig. 18). The MD simulation also reveals that the electric field is essential in stabilizing Br Frenkel defect structure in MAPB, as the defected MAPB structure is not stable under general thermodynamic conditions. Therefore, even after the withdrawal of the electric field, polarization is retained in our nanocomposite owing to the built-in electric field from poled PVDF, which stabilizes the defect structure in MAPB via interfacial coupling. This is consistent with our PFM data indicating that a large electric field is essential to induce giant electromechanical responses of the nanocomposite, and supported by a previous report that field-induced electrostriction is volatile in MAPI crystal, vanishing when the external electric field is removed[28]. It is also aligned with colossal permittivity induced in TiO$_2$ via electron-pinned defect dipoles[55]. Key to our success, however, is the presence of a ferroelectric PVDF matrix, which provides a stabilizing polar field and thus makes the substantially enhanced polarization of nanocomposites nonvolatile as seen in Fig. 2.

The strong interfacial exchange coupling between PVDF and MAPB[56], which plays a critical role in retaining the large polarization, is revealed from the enlarged images of C−F and $V^{\bullet}_{Br} - I'_{Br}$ dipoles at the interfaces indicated in Fig. 5. As seen, $V^{\bullet}_{Br} - I'_{Br}$ dipoles not only coupled much more tightly with PVDF chains but also remain largely unchanged even after the electric field was removed (Supplementary Fig. 19), leading to a large enhancement in the overall remnant polarization, consistent with the experimental observation (Fig. 2).

The finite PVDF-MAPB interface can be interpreted microscopically from the MD simulation showing width of ~2.5 nm for the 10 nm MAPB nanocrystal (Supplementary Fig. 20), which can be understood via the chemical bonding between organic cations of MAPB and fluoride atoms of PVDF (Fig. 4d), as well as phenomenlogically by Tanaka's multilayer core interfacial model[57].

## Discussion

It is quite remarkable that with less than 1% of second-phase fillers we have increased the polarization and electromechanical coupling several times, way beyond any conventional composite theories as well as previously reported composite data. Equally impressive is that the large polarization is accomplished while retaining excellent flexibility and stretchability of PVDF polymer, resulting in a champion figure of merit $P_r/Y$ for flexible ferroelectric applications. In this regard, recently reported super-elasticity in ultrathin free-standing barium titanate film[58] is accomplished mainly through the geometric softening of membrane rigidity[59], while the large $P_r/Y$ value of MAPB/PVDF nanocomposite represents the intrinsic behavior of bulk materials at the large scale, facilitating its practical flexible electronics applications. The Frenkel pair mechanism is quite intriguing, yet supported by a range of structural and functionality data as well as molecular dynamics calculation. This is also proved by the large ferroelectric polarization ($P_r$ ~ 13.6 μC cm$^{-2}$) observed in P(VDF-HFP) incorporating only 0.5% of MAPB (Supplementary Fig. 21), while pure P(VDF-HFP) exhibits typical linear dielectric characteristics. The strategy uncovered, as a result, is quite general. As a surprising byproduct of this study, we also discovered that the usually unstable MAPB is quite stable in the PVDF matrix, showing no degradation after 50 days. This understanding thus also points toward a possible direction for enhancing the stability of perovskite solar cells. We believe such hybrid perovskites–polymer nanocomposites provide a promising platform to study the interplay among optical, mechanical, and electrical properties and may lead to new multifunctional electronics.

## Methods

### Synthesis of nanocomposite films

MAPB precursor was prepared by dissolving $PbBr_2$ and MABr (1:1 molar ratio) in dimethyl formamide (DMF) solvent to make a transparent solution. Next, PVDF powder was dissolved fully after vigorously stirring for 24 h in the MAPB precursor to make MAPB/PVDF nanocomposite solution with different MAPB volume factions. The composite solution was cast onto a precleaned glass substrate at 35 °C under atmospheric conditions to evaporate the solvent. Subsequently, the films were dried in a vacuum at 40 °C for 12 h to remove the remaining trace of DMF. Finally, these flexible films were peeled from the flat substrates and dried at 40 °C overnight. The thicknesses of MAPB/PVDF and MAPB/P(VDF-HFP) composite films are around 12 μm and 15 μm, respectively.

### Structural and conformational characterization

Fourier transform infrared (Nicolet iN10MX), Raman spectroscopy (LabRAM HR Evolution) with 633 nm laser and wide-angle X-ray scattering (Xenocs Xeuss SAXS/WAXS System) measurements were employed for analyzing crystalline structure. X-ray photoelectron spectra (ESCALAB 250Xi) were performed with a monochromated Al Kα source. The absorption and photoluminescence spectra of the films were measured using an ultraviolet-visible-near infrared spectrometer (Shimadzu UV-3600) and a fluorescence spectrometer (Hitachi F-7000), respectively. Atomic force microscopy-infrared spectroscopy (NanoIR3, Bruker) was operated in contact mode, employing gold-coated Si tips (Bruker, spring constant 0.07–0.4 N m$^{-1}$), and the AFM-IR absorption spectra were obtained with IR laser irradiating at wavelength 1260 cm$^{-1}$. Differential scanning calorimetry analysis was

performed on a Thermal Analysis Instruments Analyzer (STA-449F3, NETZSCH, Germany) from 30 °C to 300 °C at a heating rate of 10 °C min$^{-1}$ in Ar atmosphere. The degree of crystallinity in the PVDF matrix was calculated from DSC curves according to the equation

$$\chi_c(\%) = \frac{\Delta H_f}{(1-\phi)\Delta H_m^{100}} \times 100\% \tag{1}$$

where $\Delta H_f$ and $\Delta H_m^{100}$ are the melting enthalpy of composite and 100% crystalline PVDF (103.4 J g$^{-1}$)[53], while $\phi$ represents the mass fraction of fillers, respectively. TEM images were acquired at JEM-2100F operated at 200 kV.

### Electrical measurements

For electrical measurements, the Cu electrodes with 2 mm in diameter were sputtered on both sides of the films. A ferroelectric analyzer (aixACCT TF-3000) equipped with a laser interferometer and high voltage amplifier was employed to measure ferroelectric behaviors, including $P$–$E$ loops, PUND, and strain-electric field curves. For the PUND test, a triangle voltage waveform was employed at a frequency of 10 Hz, where the write pulse rise time was 25 ms and the read pulse delay was 10 ms. Samples were immersed in silicon oil for high-voltage protection during the measurement. Dielectric performances of the samples were tested on Novocontrol (GmbH Concept 80) from $10^{-2}$ to $10^7$ Hz at room temperature.

Piezoelectric performances were carried out on a piezoresponse force microscope (Asylum Cypher ES). Ti/Pt-coated tip (AC240TM-R3 AFM Probe, Olympus) with a spring constant of 2 N m$^{-1}$, a tip radius of 25 nm, and a resonance frequency of 70 kHz was selected for the measurement. To carry out PFM measurements on the poled films, Cu electrodes were etched by $CuSO_4$/HCl solution after poling. A sequential excitation scanning probe microscopy (SE-SPM) technique was employed to acquire high-quality data in the frequency domain, as detailed in the Supplementary Information.

### Optical second harmonic generation measurement

The 800 nm laser (Coherent, Chameleon, 800 nm, 140 fs, 80 MHz) was used as an incident fundamental beam. The $p$-out polarized SHG signals were analyzed by a Glan prisms analyzer, and the SHG intensity was measured by a photomultiplier tube (PMT) detector.

### Mechanical property measurement

The strain-stress curves were performed on the Zwicki-Line Materials Testing Machine Z0.5 (ZwickRoell, Germany). The samples were cut into the size of 50 mm × 9 mm following the standards for tensile test and were drawn under a constant rate of 0.02 mm s$^{-1}$ till fracture.

### Molecular dynamic simulation

The molecular model of MAPB nanocrystal and PVDF nanocomposite was created by Materials Studio, and dipole moments distributions were calculated employing Large-scale Atomic/Molecular Massively Parallel Simulator package. The details are provided in Supplementary Information.

## Data availability

All data are available in the main text or the Supplementary information. The data that support this study are available from the corresponding author upon request.

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

## Acknowledgements

Y.W. acknowledges support from the National Natural Science Foundation of China (grant No. 51872009). J.L. and Y.W. acknowledge the National Natural Science Foundation of China (grant No. 92066203). J.L. acknowledges support of Shenzhen Science and Technology Innovation Committee (KQTD20170810160424889), Outstanding Talents Training Fund in Shenzhen, Guangdong Provincial Key Laboratory Program from Guangdong Science and Technology Department (2021B1212040001), P.G. thanks the support from the "2011 Program" Peking-Tsinghua-IOP Collaborative Innovation Center of Quantum Matter, and the Electron Microscopy Laboratory at Peking University.

## Author contributions

Y.W. initiated the study, and J.L. developed the idea and supervised the project. C.H. and Z.C. made the samples and characterizations under the supervision of Y.W. Z.L. performed PFM tests under the supervision of J.L., Y. Zhang performed SHG measurement, S.C. carried out TEM characterization under the supervision of P.G. Y. Zheng performed the molecular dynamic simulation under the supervision of Y.W. Y.S., J.W. and Y.D. helped with the dielectric, AFM-IR and *P–E* measurements. C.N. and C.D. helped with data analysis. Y.W. and J.L. wrote the paper with the help of all authors.

## Competing interests

The authors declare no competing interests.
