## [Peer Review File · Nature Communications]

Halide Perovskite Inducing Anomalous Nonvolatile Polarization in Poly(vinylidene fluoride)-based Flexible NanocompositesREVIEWER COMMENTS

Reviewer #1 (Remarks to the Author):

PVDF and MAPbBr₃ nanocomposites were reported to enhance the polarization of PVDF from 10 μC to 19.6 $\mu\text{C cm}^{-2}$. The authors still claimed them to be ferroelectrics, though they understand that the polarization is not contributed by traditional mechanisms in ferroelectric materials. In this case, the apparent polarization is more likely from metal ions in MAPbBr₃ or defects. I am not sure it can be called ferroelectric material. Only P-E loops were shown by they are known to be misleading, because of many issues/artifacts with such measurement. I donot think it is counterintuitive, because one would naturally predict these behaviors. It is not clear to this reviewer whether this represents a new set of materials, nor they can be useful.

Reviewer #3 (Remarks to the Author):

The manuscript reports a kind of ferroelectric nanocomposite, PVDF with 0.94% of self-nucleated CH₃NH₃PbBr₃(MAPB) nanocrystals, and demonstrate its polarization performance, as well as discussed, the mechanism. The MAPB/PVDF nanocomposites have large polarization as well as good stretchability. This manuscript is recommended to be published after minor revision.

(1)It is demonstrated that the key to the synthesis is simultaneous nucleation of MAPB nanocrystal confined within polymer chains, which ensuring the overcome of the major difficulties. Is there any evidence for the process, such as experimental or simulation investigation?

(2)It is suggested to provide the UV-vis absorption and photoluminescence spectra of MAPB in Fig. 1e for comparison.

(3)It is demonstrated that the new peaks in FTIR spectra provide direct evidence for a strong interfacial coupling strengthened by poling-induced Frenkel pairs, and shift of F 1s X-ray photoelectron spectra to electron transfer from fluorine atom to MAPB induced by high voltage poling. But the variation in the spectra is not clear enough for convincing. Please provide more evidence.

(4)The authors discussed that the built-in electric field from poled PVDF can retained the polarization in the nanocomposite from the Br Frenkel defect structure in MAPB. Please provide the XPS spectra of MAPB before and after poling to further support the conclusion.

(5)There are some mistakes in the format. Please check it and revise the manuscript carefully.

Reviewer #4 (Remarks to the Author):

In this manuscript, the authors incorporated CH₃NH₃PbBr₃ (MPAB) in PVDF matrix to enhance the polarization of PVDF by almost threefold while retaining its intrinsic flexibility and stretchability. This work provides an innovative solution to enhance properties and multifunctionality of polymeric ferroelectrics, and it is an interesting work for the ferroelectric community. Thus, I recommend this article for publication in Nature Communications. However, some issues regarding the discussion on the crystallinity and stability of the MAPB/PVDF nanocomposite should be further clarified.

1.What is the mechanism of the cohesive binding between MAPB and PVDF?

2.As the previous reference reported, the phases can effectively affect the polarity and

ferroelectric properties of PVDF, and the β -phase PVDF is easily polarized under an electric field (Adv. Mater. 2022, 34, 2110482; Nat. Protoc. 2018, 13, 681). Thus, crystalline structure of the PVDF film with or without MAPB should be carefully demonstrated and analyzed.

3. In page 13, the authors claimed that both the β -phase PVDF and MAPB showed compressive strain in the poled nanocomposite film. What is the reason for this phenomenon?

4. In this manuscript, the enhanced polarizability of the PVDF was attributed to the defect dipoles in MAPB. Would the crystalline structure of MAPB be changed after poling?

5. The MA based hybrid perovskite is sensitive to water, oxygen, light and heat. However, the authors claimed that "The MAPB/PVDF nanocomposite film exhibits excellent polarization stability", how is it possible? Authors should comment on this issue.

6. How about the mechanical stability of the nanocomposite film? The polarizability of the nanocomposite film versus stretching or bending cycles should be provided.

Response to Reviewers' Comments

We appreciate the valuable and constructive comments from three referees that help us to improve the manuscript further. In response to these comments, we have carried out extensive additional experiments to address the issues and concerns raised, and we have made thorough revisions correspondingly, as detailed in our point-to-point responses to the referees below. Key revisions are also highlight in blue in the manuscript.

Reviewer #1

Comments:

PVDF and MAPbBr₃ nanocomposites were reported to enhance the polarization of PVDF from 10 $\mu\text{C cm}^{-2}$ to 19.6 $\mu\text{C cm}^{-2}$. The authors still claimed them to be ferroelectrics, though they understand that the polarization is not contributed by traditional mechanisms in ferroelectric materials. In this case, the apparent polarization is more likely from metal ions in MAPbBr₃ or defects. I am not sure it can be called ferroelectric material. Only P-E loops were shown by they are known to be misleading, because of many issues/artifacts with such measurement. I do not think it is counterintuitive, because one would naturally predict these behaviors. It is not clear to this reviewer whether this represents a new set of materials, nor they can be useful.

Response:

We thank the reviewer for raising these important issues. Ferroelectricity is understood as a long-range polar order that breaks crystalline symmetry, and this polar order can be switched by external field. In addition to conventional **P-E measurements** as the referee mentioned, in the original manuscript we have also included **PUND measurements** to mitigate artifacts, as well as **PFM measurements** at microscopic scale to support our conclusion. Nevertheless, we understand the referee's concern, and it is important to clarify it. As a result, we carried out additional **second harmonic generation (SHG) measurement**, which is not affected by artifacts such as injected charges and leakage currents, to further support our conclusion, as shown in **Fig. R1**. It is evident that poled nanocomposites exhibits much stronger SHG intensity compared

to poled PVDF (Fig. R1a,b), consistent with our P-E, PUND, and PFM data, while both as processed nanocomposites and PVDF before poling exhibit negligible SHG response (Fig. R1c), suggesting poling is essential to establish long-range macroscopic polar order, consistent with our PFM data. In other words, there is long-range polar order in the nanocomposite that can be switched by the external electric field, and thus we believe it is appropriate to call it ferroelectric, regardless its microscopic origin.

Fig. R1 Comparison of SHG intensities generated from both poled and as-prepared MAPB/PVDF and PVDF films, respectively. **a** Polar plots of transmitted SHG intensity for poled films with the solid lines fitting based on an orthogonal $mm2$ polar structure. **b** Multiple points statistics on SHG intensities. **c** Polar plots of transmitted SHG intensities generated from as-prepared PVDF and MAPB/PVDF films with the insets showing enlarged SHG signals with very low intensity.

Whether it is *counterintuitive* or not is quite subjective, and thus we decide to remove it from the manuscript. Nevertheless, we believe MAPB/PVDF nanocomposite is *new*, and it possesses multiple functionalities, including ferroelectricity, electromechanical coupling, photoluminescence, and mechanical flexibility. We believe these make it potentially useful, which we are currently exploring.

In the revision, we have incorporated **Fig. R1a,b** as the revised **Fig. 2c**, and **Fig. R1c** as revised **Fig. S8b** in the SI, with the following discussions on page 10:

“To further confirm that the large polarization indeed arises from long-range polar order breaking the crystalline symmetry, optical second harmonic generation (SHG) technique has been employed⁴⁷ (Supplementary Fig. 8a). Transmitted SHG signals generated from both poled MAPB/PVDF and PVDF films with respect to incident light polarization angle are presented in Fig. 2c, in sharp contrast to the as-prepared films showing negligible SHG responses (Supplementary Fig. 8b). The *p*-out SHG polar plots of both films display a twofold symmetry, which is well fitted by an orthogonal *mm*2 structure (Supplementary Information, Section 7). Notice that the average SHG intensity of poled MAPB/PVDF is twice the value of poled PVDF (Fig. 2c, Supplementary Fig. 8c), indicating substantially enhanced remnant polarization P_r in MAPB/PVDF nanocomposite, consistent with the *P-E* and PUND results.”

Reviewer #3

Comment:

The manuscript reports a kind of ferroelectric nanocomposite, PVDF with 0.94% of self-nucleated CH₃NH₃PbBr₃(MAPB) nanocrystals, and demonstrate its polarization performance, as well as discussed, the mechanism. The MAPB/PVDF nanocomposites have large polarization as well as good stretchability. This manuscript is recommended to be published after minor revision.

Response:

We thank the reviewer for the positive comments and recommendation.

Comment:

(1) It is demonstrated that the key to the synthesis is simultaneous nucleation of MAPB nanocrystal confined within polymer chains, which ensuring the overcome of the major difficulties. Is there any evidence for the process, such as experimental or simulation investigation?

Response:

We thank the reviewer for the advice, and as suggested, we recorded synthesis process, and observed rather distinctive transformation judged from color and macroscopic appearance variation, as shown in **Fig. R2**. When the mixture solution was cast onto the glass substrate at 40 °C, the DMF solvent gradually evaporated, and the solid polymer film began to form. Subsequently, the yellow-green color appeared from the corner of the film, indicating the initial nucleation of MAPB nanocrystals, and the color gradually spread to the center, and the whole film turned to yellow-green color in another 20 min, completing the formation of nanocomposite film. This suggests that nucleation of MAPB nanocrystals and formation of solid PVDF matrix is more or less simultaneous, though the nucleation process is inhomogeneous, starting at the corner.

Fig. R2 Pictures recorded from the film formation process showing nucleation of MAPB nanocrystals and PVDF film.

In the revision, we have added **Supplementary Video 1** and included Fig. R2 as **Fig. S1a** in the SI, with the following discussions on page 5-6 in the main text:

“The key to our synthesis is that nucleation of MAPB nanocrystals and formation of PVDF polymer matrix is almost simultaneous (Supplementary Video 1 and Supplementary Fig. 1a), ensuring well dispersion of MAPB nanocrystals inside PVDF without crystal growth and agglomeration.”

Comment:

(2) It is suggested to provide the UV-vis absorption and photoluminescence spectra of MAPB in Fig. 1e for comparison.

Response:

As suggested, we have supplemented the UV-vis absorption and photoluminescence spectra of the MAPB nanoparticles dispersed in n-hexane solvent for comparison as shown below in **Fig. R3**. The MAPB/PVDF nanocomposite films show similar light absorption and photoluminescent properties as those of MAPB nanoparticles with similar size. Due to limited space in Fig. 1e, in the revision we added UV-vis absorption and PL spectra of MAPB nanoparticles as **Fig. S4** in the SI, with the following discussions on page 7 in the main text:

“The absorption onset of MAPB/PVDF nanocomposite occurs at around 540 nm, while the photoluminescence (PL) emission peak occurs at around 530 nm with full width at half maximum 24.4 nm, consistent with the band-to-band transition of bromide perovskite³³ and those measured from colloidal MAPB nanoparticles (Supplementary Fig. 4).”

Fig. R3 UV-vis absorption and photoluminescence spectra of MAPB nanoparticles in n-hexane solvent with the inset showing the photo of the MAPB NPs dispersion and their morphology observed by TEM.

Comment:

(3) It is demonstrated that the new peaks in FTIR spectra provide direct evidence for a strong interfacial coupling strengthened by poling-induced Frenkel pairs, and shift of F 1s X-ray photoelectron spectra to electron transfer from fluorine atom to MAPB induced by high voltage poling. But the variation in the spectra is not clear enough for convincing. Please provide more evidence.

Response:

We thank the reviewer for the valuable suggestion, and we examined more data to verify the conclusion regarding interfacial coupling strengthened by poling. In **Fig. R4**, poled MAPB/PVDF film shows slightly reduced optical bandgap of MAPB and photoluminescent peak shifting to higher wavelength, consistent with the conclusion that poling generates more defect states. The Raman spectra (**Fig. R5**) reveal that MAPB/PVDF nanocomposite retains γ -phase (Fig. R5b, c) after poling, whereas in PVDF the chain stacking in a portion of γ -phase has been rotated by poling as evidenced by the sharp decline in vibration peak intensity at 812 cm^{-1} for γ -phase (Fig. R5a, c), again suggesting strong interfacial coupling in the nanocomposite. Such interfacial coupling is also supported by MD calculations shown in Fig. 5 and Supplementary Fig. 17.

Fig. R4 Poling induced variation in optical properties. a UV-vis absorption, and **b** PL spectra of MAPB/PVDF nanocomposite films before and after poling.

Fig. R5 Poling induced variation in phase structure. Comparison on Raman spectra of **a** PVDF, **b** MAPB/PVDF before and after poling and **c** poled MAPB/PVDF and PVDF films.

In the revision, we have added Fig. R4 as Fig. S14, Fig. R5 as Fig. S13 in the SI, with the following discussions on page 15-16 in the main text:

“Raman spectra reveal that MAPB/PVDF nanocomposite retains γ -phase after poling (Supplementary Fig. 13b, c) in contrast to PVDF wherein γ -phase is partially transformed by poling (Supplementary Fig. 13a, c), further demonstrating the strong interfacial coupling between PVDF and MAPB.”

and

“The sharp decline in intensities for Br 3d and N 1s XPS signals after poling together with the slightly reduced optical bandgap of MAPB and shifting in PL peak to higher wavelength (Supplementary Fig. 14) also support the formation of Br vacancies in poled MAPB, generating extra positive charges and thus stronger bonding with F atoms in PVDF, favoring an ordered arrangement of dipoles toward poling direction.”

Comment:

(4) The authors discussed that the built-in electric field from poled PVDF can retained the polarization in the nanocomposite from the Br Frenkel defect structure in MAPB. Please provide the XPS spectra of MAPB before and after poling to further support the conclusion.

Response:

As suggested, comparison on XPS spectra of Br and N elements from MAPB before and after poling has been presented in **Fig. 4d** to support the conclusion that Br Frenkel defect structure has been created by high electric field poling. In the revision, we also

added the following discussions on page 16 in the main text:

“The sharp decline in intensities for Br 3d and N 1s XPS signals after poling together with the slightly reduced optical bandgap of MAPB and shifting in PL peak to higher wavelength (Supplementary Fig. 14) also support the formation of Br vacancies in poled MAPB, generating extra positive charges and thus stronger bonding with F atoms in PVDF, favoring an ordered arrangement of dipoles toward poling direction.”

Comments:

(5) There are some mistakes in the format. Please check it and revise the manuscript carefully.

Response:

We thank the reviewer for the careful reading on our manuscript. We have revised the manuscript thoroughly to fix the format errors.

Reviewer #4

In this manuscript, the authors incorporated $\text{CH}_3\text{NH}_3\text{PbBr}_3$ (MPAB) in PVDF matrix to enhance the polarization of PVDF by almost threefold while retaining its intrinsic flexibility and stretchability. This work provides an innovative solution to enhance properties and multifunctionality of polymeric ferroelectrics, and it is an interesting work for the ferroelectric community. Thus, I recommend this article for publication in Nature Communications. However, some issues regarding the discussion on the crystallinity and stability of the MAPB/PVDF nanocomposite should be further clarified.

Response:

We thank the reviewer for the positive comments and recommendation.

Comment:

1. What is the mechanism of the cohesive binding between MAPB and PVDF?

Response:

We ascribe the cohesive binding between MAPB and PVDF to the following aspects: (1) Highly polar C-F bond; (2) Unique organic-inorganic hybrid ABX_3 molecular structure; (3) Confined nucleation of MAPB within PVDF chains. F atom has the strongest electronegativity and is inclined to take electrons from surrounding atoms with low electronegativity, establishing strong molecular interaction. MAPB contains methylamine cations, which are prone to form bond interaction with C-F. Thanks to the confined nucleation process of MAPB/PVDF nanocomposite, the nucleation of MAPB nanocrystals is confined within the free volume of PVDF chains. Along with the evaporation of the solvent, the crystallization of PVDF causes the shrink of PVDF chains and impedes mass transfer of solute ions, thus limits the crystal growth of MAPB and results in tight binding between PVDF and MAPB. Therefore, cohesive binding between MAPB and PVDF has been realized.

In the revision, we also added the following discussions on page 6 in the main text to address the issue:

“It also results in cohesive binding between MAPB and PVDF matrix, overcoming major difficulties associated with processing of PVDF-based nanocomposites³², especially those involving unstable materials such as halide perovskites, thanks to the strong molecular interaction enabled by highly polar C-F bond with strong electronegativity and unique hybrid \$ABX_3\$ molecular structure containing methylamine cations.”

Comment:

2. As the previous reference reported, the phases can effectively affect the polarity and ferroelectric properties of PVDF, and the β -phase PVDF is easily polarized under an electric field (Adv. Mater. 2022, 34, 2110482; Nat. Protoc. 2018, 13, 681). Thus, crystalline structure of the PVDF film with or without MAPB should be carefully demonstrated and analyzed.

Response:

We thank the reviewer for the good suggestion. Crystalline structure of the PVDF film with or without MAPB has been carefully analyzed and demonstrated via multiple characterization methods, including WAXS, FTIR, Raman and DSC. We can confirm that the crystalline structure of the PVDF film fabricated here is polar, containing both β and γ phases. Compositing PVDF with MAPB does not alter the phase structure of PVDF or increase the content of β -phase much as evidenced from FTIR (Fig. 1d), 1D-WAXS (**Fig. R6a**), Raman spectra (**Fig. R6b**). Poling indeed promotes the polarization of β -phase in PVDF with or without MAPB as previously reported^[R1,R2], which is significant for PVDF as evidenced from the largely increased intensities of β -phase (110)/(200) diffraction peak (**Fig. R7**), FTIR (Fig. 4c, the intensified peak at 1275 cm^{-1} for all-*trans* chains) and Raman spectra (**Fig. R5a**) but to less extent in the MAPB/PVDF nanocomposite as shown from 1D-WAXS (Fig. 4a), FTIR (Fig. 4c) and Raman spectra (Fig. R5b, c). Despite the poling induced variation in polarized β -phase observed in both PVDF and MAPB/PVDF films, the variation in γ -phase has also been revealed from Raman spectra. MAPB/PVDF nanocomposite retains γ -phase after poling (Fig. R5b, c), whereas the chain packing of a portion of γ -phase in PVDF has been rotated by poling as evidenced by the sharp decline in vibration peak intensity at 812 cm^{-1} for the γ -phase (Fig. R5a, c), indicating that MAPB nanoparticles constrain the chain movement to some extent in MAPB/PVDF via interfacial coupling.

Fig. R5 Poling induced variation in phase structure. Comparison on Raman spectra of **a** PVDF, **b** MAPB/PVDF before and after poling and **c** poled MAPB/PVDF and PVDF films.

Fig. R6 Crystalline structure of as-prepared MAPB/PVDF in comparison with PVDF. a 1D WAXS patterns, **b** Raman spectra.

Fig. R7. Poling induced variation in crystalline structure of PVDF film. 1D-WAXS patterns of PVDF film before and after poling.

In the revision, we added Fig. R5 as Fig. S13, Fig. R6 as Fig. S3 and Fig. R7 as Fig. S12 in SI, and cited Refs. R1, R2 as Refs 52, 53, with the relevant discussion added to the main text on page 6:

“Fourier transform infrared spectra (FTIR) (Fig. 1d) indicates that there is no detectable phase transition involved in PVDF induced by MAPB, consistent with the phase analysis via X-ray and Raman spectra (Supplementary Fig. 3).”

On page 15:

“Phase structure variation analyses via combined WAXS, FTIR and Raman spectra show that poling enhances polarized β -phase more significantly in PVDF than that of MAPB/PVDF film (Fig. 4a, c and Supplementary Figs. 12, 13), which further support

that polarized β -phase cannot accounts for the observed polarization enhancement.”

and

“Raman spectra reveal that MAPB/PVDF nanocomposite retains γ -phase after poling (Supplementary Fig. 13b, c) in contrast to PVDF wherein γ -phase is partially transformed by poling (Supplementary Fig. 13a, c), further demonstrating the strong interfacial coupling between PVDF and MAPB.”

Comment:

3. In page 13, the authors claimed that both the β -phase PVDF and MAPB showed compressive strain in the poled nanocomposite film. What is the reason for this phenomenon?

Response:

For the PVDF, the compressive strain is attributed to its negative longitudinal piezoelectric coefficient. When applied electric field vertical to the thickness direction of the film, contraction in film thickness along the same direction will take place, causing compressive strain^[R3]. For the MAPB, the compressive strain is attributed to electrostriction originated from lattice deformation due to formation of additional defects under applied bias, as elaborately discussed in the work by Huang et al^[R4].

In the revision, we also added the following discussions on page 14-15 in the main text:

“...generating a large compressive strain of -1.9% due to its negative longitudinal piezoelectric coefficient⁴⁴ for the poled β -phase PVDF and a compression of -0.5% for MAPB nanocrystals. The electrostrictive behavior of MAPB nanocrystals within PVDF matrix is consistent with the observed large electrostriction in lead halide perovskites single crystals, where Frenkel dipoles were generated and stabilized by electric field, accompanied by a lattice contraction²⁸.”

Comment:

4. In this manuscript, the enhanced polarizability of the PVDF was attributed to the defect dipoles in MAPB. Would the crystalline structure of MAPB be changed after

poling?

Response:

There is indeed deformation in crystal lattice of MPAB after poling with defect dipoles generated, where the original cubic structure ($a = 5.97 \text{ \AA}$) has contracted with a reduced lattice constant of 5.93 \AA , as calculated from 1D-WAXS pattern (Fig. 4a). According to the DFT study on electrostrictive behavior of halide perovskite by Huang^[R4], when the Frenkel dipoles were generated and stabilized by electric field, these defects migrated and separated along with electric field direction, accompanied by a lattice contraction. This consistent with our experimental observation.

In the revision, we also added the following discussions on page 15 in the main text:

“The electrostrictive behavior of MAPB nanocrystals within PVDF matrix is consistent with the observed large electrostriction in lead halide perovskites single crystals, where Frenkel dipoles were generated and stabilized by electric field, accompanied by a lattice contraction²⁸.”

Comment:

5. The MA based hybrid perovskite is sensitive to water, oxygen, light and heat. However, the authors claimed that “The MAPB/PVDF nanocomposite film exhibits excellent polarization stability”, how is it possible? Authors should comment on this issue.

Response:

Indeed, MA based hybrid halide perovskites are sensitive to water, oxygen, light and heat. However, in our case, MAPB nanocrystals with size in order of 10 nanometers are well monodispersed in and encapsulated by PVDF matrix (Fig. 1b), which is quite inert, making the nanocomposites stable. Furthermore, the presence of the strong C-F bond has a crucial impact on the high resistance to oxidation and to hydrolytic stability^[R5]. We also added new stability test as **Fig. R8**, showing that the nanocomposites hardly changed its photoluminescence property after 10 days in water, to further support the conclusion.

Fig. R8 PL spectra of MAPB/PVDF film before and after soaking in water for 10 days.

In the revision, we included Fig. R8 as Fig. S6 in the SI, and also added the following discussions on page 9 in the main text:

“The photoluminescence property of the nanocomposite is also quite stable, hardly changes after 10 days in water (Supplementary Fig. 6). Such excellent stability is quite unusual for halide perovskites. The key may lie in the well monodispersed MAPB nanocrystals in PVDF matrix, whose strong C-F bond has a crucial impact on the high resistance to oxidation and to hydrolytic stability¹², thus effectively protecting MAPB nanocrystals from decomposition.”

Comment:

6. How about the mechanical stability of the nanocomposite film? The polarizability of the nanocomposite film versus stretching or bending cycles should be provided.

Response:

As suggested, we have carried out P-E loop test on stretched film via Materials Testing Machine under constant rate of 0.02 mm/s. As seen from **Fig. R9**, the *P-E* loop is stable, insensitive to deformation as high as 5%.

Fig. R9 Mechanical stability test on MAPB/PVDF nanocomposite film. *P-E* hysteresis loops of MAPB/PVDF film after stretched at strain 2% and 5%, respectively.

In the revision, we have added Fig. R9 as Fig. S10 in the SI, with the relevant discussion in the main text on Page 11:

“Such large polarization is also quite stable under stretch. The *P-E* loop is insensitive to deformation as high as 5% (Supplementary Fig. 10).”

References

- [R1] Ribeiro, C. et al. Electroactive poly(vinylidene fluoride)-based structures for advanced applications. *Nat. Protoc.* **13**, 681-704 (2018).
- [R2] Chen, W. et al. High-polarizability organic ferroelectric materials doping for enhancing the built-in electric field of perovskite solar cells realizing efficiency over 24%. *Adv. Mater.* **34**, 2110482 (2022).
- [R3] Katsouras, I. et al. The negative piezoelectric effect of the ferroelectric polymer poly(vinylidene fluoride). *Nat. Mater.* **15**, 72-84 (2016).
- [R4] Chen, B. et al. Large electrostrictive response in lead halide perovskites. *Nat. Mater.* **17**, 1020-1026 (2018).
- [R5] Ameduri, B. From Vinylidene fluoride (VDF) to the applications of VDF-containing polymers and copolymers: recent developments and future trends. *Chem. Rev.* **109**, 6632-6686 (2009).

REVIEWER COMMENTS

Reviewer #1 (Remarks to the Author):

Prof Li is an expert in ferroelectric materials. This is some interest discovery, and may be published in Nature Communication, but I hope it is not overclaimed.

I still don't understand underlying microscopic mechanism for the enhanced polarization despite of the claims. What can provide a higher density of charges than the already very high ones from PVDF, or it is just ion migration? I do notice that the measurement frequency is very low.

For the claim of using the nanocomposite for both ferroelectrics and optic-active, I would want to know what application would need it. And when charges are generated by strong light, will it still a good ferroelectric material? How about leakage current and screening by light generated charges?

Reviewer #3 (Remarks to the Author):

The manuscript reports the incorporation of $\text{CH}_3\text{NH}_3\text{PbBr}_3$ (MAPB) nanocrystals into PVDF matrix to obtain $\sim 19.6 \text{ 35 C/cm}^2$ polarization with 0.94% of MAPB and investigates the mechanism that the enhancement is accomplished via electric field induced Frenkel pairs in MAPB stabilized by the poled PVDF through interfacial coupling. This work is meaningful. Therefore, it is recommended to be published after minor revision.

(1) In this work, the authors proposed a strategy of using interface coupling between MAPB and PVDF to improve the polarization. However, many previous literatures have reported similar methods to improve PVDF performances. Compared with those reported literature, what is the advance and originality of MAPB/PVDF? It is suggested that the authors can further refine the distinguishment to emphasize the importance of this work over the reported literature.

(2) Please further elaborate on how the phenomena in Video 1 and Supplementary Fig. 1a indicate that the nucleation of MAPB nanocrystals and formation of PVDF polymer matrix is almost simultaneous.

(3) The authors indicates that it is observed from the TEM image in Fig. 1b that the MAPB nanocrystals is monodisperse in PVDF matrix. However, it is difficult to be convincing. The author should provide more evidence. In addition, please explain why the MAPB can form monodisperse during growth rather than small aggregation.

(4) The authors reported that the cohesive binding between MAPB and PVDF matrix is owing to the strong molecular interaction enabled by highly polar C-F bond with strong electronegativity and unique hybrid ABX_3 molecular structure containing methylamine cations, which is important to the enhancement of polarization of MAPB/PVDF. However, there is no sufficient evidence. Please provide more evidence and sufficient analysis.

(5) It is obvious that the MAPB content in MAPB/PVDF has a significant effect on its performance. It is suggested to study the effect of MAPB content on residue polarization, stability and mechanical properties, as well as its response mechanism, which is one of the most concerned problems of researchers.

Reviewer #4 (Remarks to the Author):

All my questions have been clearly addressed by the authors. I think this is a high quality paper and can be published as is now.

Response to Reviewers' Comments

We appreciate the valuable and constructive comments from three referees that help us to improve the manuscript further. In response to these comments, we have carried out additional experiments to address the issues and concerns raised, and we have made thorough revisions correspondingly, as detailed in our point-to-point responses to the referees below. Key revisions are also highlight in blue in the manuscript.

Reviewer #1

Comment:

I still do not understand underling microscopic mechanism for the enhanced polarization despite of the claims. What can provide a higher density of charges than the already very high ones from PVDF, or it is just ion migration? I do notice that the measurement frequency is very low.

Response:

We appreciate this important question, and we hope to clarify the underlying mechanism as follows:

- 1) Poling electric field induces Frenkel pairs in MAPB, resulting in large defect dipoles that is arranged in ferroelectric order. This microscopic interpretation is supported by TEM image (**Fig. 4b**) and MD simulations (**Fig. 5c**).
- 2) The defect dipole of MAPB is stabilized by the depolarization field of PVDF matrix after the removal of poling field, and the strong interfacial interaction between MAPB and PVDF resulting in slight conformational variation of PVDF, as indicated by the new FTIR peak at 1260 cm^{-1} (**Fig. 4c and Supplementary Fig. 15**) and extensive structure data presented in the manuscript.

The Frenkel pairs are indeed induced by field driven ion migration, but the migration is local, resulting in ordered defect dipoles. It does not damage MAPB, as indicated by the strong optic-activity of MAPB/PVDF. Such defect dipoles have also been reported by previous groups and in other materials such as TiO_2 , and it is also known to have slower frequency response.

In short, large defect dipoles of MAPB and strong interfacial interaction between MAPB and PVDF are responsible for the enhanced polarization. We highlight such mechanism in the Abstract:

“Comprehensive analysis suggests that the enhancement is accomplished via delicate defect engineering, with electric field-induced Frenkel pairs in MAPB stabilized by the poled PVDF through interfacial coupling.”

And on page 4:

“A more deliberate contemplation, however, suggests the ionic migration that is responsible for their instability^{26,27} also induces defect dipoles in materials²⁸, and thus it may lead to dipolar order under appropriate tuning given its excellent polarizability. The key thus lies in making such defect dipoles nonvolatile, which we believe can be stabilized by the PVDF matrix, and this is essentially our strategy.”

Comment:

For the claim of using the nanocomposite for both ferroelectrics and optic-active, I would want to know what application would need it. And when charges are generated by strong light, will it still a good ferroelectric material? How about leakage current and screening by light generated charges?

Response:

For the potential applications, and possible interactions between light and polarization (such as how photo-induced charges and leakage current affect ferroelectric polarization), we add the following discussion on page 7-8:

“Such photoactivity may be used for anti-counterfeiting ferroelectric memory, and may even enable novel multi-bit data storage. Additional applications as well as new phenomena may also emerge from interplay between polarization and light, which is currently under our investigation.”

Reviewer #3

Comment:

The manuscript reports the incorporation of $\text{CH}_3\text{NH}_3\text{PbBr}_3$ (MAPB) nanocrystals into PVDF matrix to obtain $\sim 19.6 \mu\text{C}/\text{cm}^2$ polarization with 0.94% of MAPB and investigates the mechanism that the enhancement is accomplished via electric field induced Frenkel pairs in MAPB stabilized by the poled PVDF through interfacial coupling. This work is meaningful. Therefore, it is recommended to be published after minor revision.

Response:

We thank the reviewer for the recommendation and encouragement.

Comment:

(1) In this work, the authors proposed a strategy of using interface coupling between MAPB and PVDF to improve the polarization. However, many previous literatures have reported similar methods to improve PVDF performances. Compared with those reported literature, what is the advance and originality of MAPB/PVDF? It is suggested that the authors can further refine the distinguishment to emphasize the importance of this work over the reported literature.

Response:

As suggested, we highlight the advance and originality of MAPB/PVDF on page 4:

“While composite approach incorporating large volume fraction of ferroelectric oxide fillers have been widely used to enhance the electromechanical properties of polymers¹⁷, it also increases stiffness substantially while decreases stretchability and breakdown strength¹⁸, defeating the very purpose of polymeric ferroelectrics. We solve this problem by using less than 1% volume fraction of MPAB self-nucleated in PVDF matrix.”

And on page 19:

“It is quite remarkable that with less than 1% of second phase fillers we have increased the polarization and electromechanical coupling by several times, way beyond any conventional composite theories as well as previously reported composite data. Equally

impressive is that the large polarization is accomplished while retaining excellent flexibility and stretchability of PVDF polymer, resulting in a champion figure of merit P_r/Y for flexible ferroelectric applications.”

Comment:

(2) Please further elaborate on how the phenomena in Video 1 and Supplementary Fig. 1a indicate that the nucleation of MAPB nanocrystals and formation of PVDF polymer matrix is almost simultaneous.

Response:

We thank the reviewer for the comment, and to make it clearer, we have carried out *in situ* PL spectra measurement (**Supplementary Fig. 1b**) during the film formation process to monitor the dynamic nucleation of MAPB nanocrystals, along with the following discussion on page 6:

“The key to our synthesis is that nucleation of MAPB nanocrystals and formation of PVDF polymer matrix is almost simultaneous, as revealed by Supplementary Video 1 and Supplementary Fig. 1a, b, wherein crystallization of PVDF starts before 20 minute as indicated by gradual loss of transparency of as-cast film, while nucleation of MAPB nanocrystals completes at about 20 minute, as indicated by the stable photoluminescence intensity.”

Supplementary Fig. 1 Formation of MAPB/PVDF nanocomposite film. **a** Pictures recorded from the film formation process showing nucleation of MAPB nanocrystals and PVDF film. **b** *in situ* PL spectra of MAPB/PVDF nanocomposite film during film formation process. **c** Photo of a highly homogenous piece of nanocomposite film rolled around a cylinder.

Comment:

(3) The authors indicate that it is observed from the TEM image in Fig.1b that the MAPB nanocrystals is monodisperse in PVDF matrix. However, it is difficult to be convincing. The author should provide more evidence. In addition, please explain why the MAPB can form monodisperse during growth rather than small aggregation.

Response:

We thank the reviewer for the comment. As suggested, we have obtained additional TEM images to demonstrate the microstructures of MAPB/PVDF nanocomposite, and added the following discussion on page 6:

“As confirmed by the low magnification transmission electron microscopy (TEM, Fig.1b), the microstructure of nanocomposite film shows monodisperse MAPB nanocrystals in PVDF matrix with average particle diameter of 9.8 nm, which is further

supported by higher magnification TEM as well as high resolution TEM (HRTEM) images in Supplementary Fig. 2.”

Supplementary Fig. 2 a TEM image of MAPB/PVDF film. b, c HRTEM images of two well separated MAPB nanocrystals taken from different regions.

We also added discussion on why the MAPB can form monodisperse during growth rather than small aggregation on page 6:

“...wherein crystallization of PVDF starts before 20 minute as indicated by gradual loss of transparency of as-cast film, while nucleation of MAPB nanocrystals completes at about 20 minute, as indicated by the stable photoluminescence intensity. This ensures well dispersion of MAPB nanocrystals inside PVDF, since the free volume of PVDF polymer provides the space for MAPB nucleation while the simultaneous film formation confines further crystal growth as well as agglomeration.”

Comment:

(4) The authors reported that the cohesive binding between MAPB and PVDF matrix is owing to the strong molecular interaction enabled by highly polar C-F bond with strong electronegativity and unique hybrid ABX₃ molecular structure containing methylamine cations, which is important to the enhancement of polarization of MAPB/PVDF. However, there is no sufficient evidence. Please provide more evidence and sufficient analysis.

Response:

As suggested, we carried out additional AFM-IR experiment, and added the following

discussions on page 16:

“FTIR spectra reveal that a new vibrating mode has emerged in MAPB/PVDF nanocomposites after poling (Fig. 4c), which is absent in both poled PVDF and unpoled MAPB/PVDF nanocomposite, providing direct evidence for a strong interfacial coupling strengthened by poling-induced Frenkel pairs. This is further supported by atomic force microscopy-infrared spectroscopy (AFM-IR) mappings (Supplementary Fig. 15) showing strong IR responses to 1260 cm^{-1} irradiation in poled MAPB/PVDF, yet no response in either poled PVDF or unpoled MAPB/PVDF.”

Supplementary Fig. 15 Simultaneously obtained topography and AFM-IR chemical maps of MAPB/PVDF nanocomposite in comparison with PVDF film with IR laser irradiating at 1260 cm^{-1} . **a** PVDF, **b** poled PVDF, **c** MAPB/PVDF and **d** poled MAPB/PVDF. The scanning area is $2\text{ }\mu\text{m} \times 2\text{ }\mu\text{m}$.

We also highlight the strong interfacial interactions on page 16:

“Shift of F 1s X-ray photoelectron spectra (XPS) to higher binding energy (Fig. 4d) in the MAPB/PVDF nanocomposite is observed, which is absent in PVDF after poling, suggesting electron transfer from fluorine atom to MAPB induced by high voltage poling. The sharp decline in intensities for Br 3d and N 1s XPS signals after poling together with the slightly reduced optical bandgap of MAPB and shifting in PL peak to higher wavelength (Supplementary Fig. 16) also support the formation of Br vacancies in poled MAPB, generating extra positive charges and thus stronger bonding with F

atoms in PVDF, favoring an ordered arrangement of dipoles toward poling direction.”

Comment:

(5) It is obvious that the MAPB content in MAPB/PVDF has a significant effect on its performance. It is suggested to study the effect of MAPB content on residue polarization, stability and mechanical properties, as well as its response mechanism, which is one of the most concerned problems of researchers.

Response:

We thank the reviewer for the suggestion. We have actually made a series MAPB/PVDF with increasing MAPB nanocrystal content. As MAPB content exceeds 0.94%, the film became leaky. Therefore, nanocomposites with MAPB content of 0.47% and 0.94% are reported here.

-As shown in **Fig. 2a**, the remanent polarization (P_r) increases from 7.3 $\mu\text{C}/\text{cm}^2$ for PVDF to 12.6 $\mu\text{C}/\text{cm}^2$ for 0.47% MAPB/PVDF and 19.6 $\mu\text{C}/\text{cm}^2$ for 0.94% MAPB/PVDF film.

-The electrical, mechanical and optical performances of MAPB/PVDF are quite stable. As presented in the **Supplementary Information**, the polarization remains 90% after placed in ambient environment (**Supplementary Fig. 6**), the photoluminescence remains unchanged after soaking in water for 10 days (**Supplementary Fig. 7**), and the large polarization is quite stable under mechanical deformation, i.e., 5% stretch (**Supplementary Fig. 11**).

-Since modulus of halide perovskite is order of magnitude smaller than typical inorganic materials, incorporation of tiny fraction of MAPB nanocrystals here, i.e., 0.47% and 0.94%, does not change the mechanical properties.

-From all the comparisons above, the slight variation in MAPB content does not appear to change the response mechanism of MAPB/PVDF.

Reviewer #4

All my questions have been clearly addressed by the authors. I think this is a high quality paper and can be published as is now

Response:

We sincerely appreciate the reviewer for the recommendation.

REVIEWERS' COMMENTS

Reviewer #1 (Remarks to the Author):

I donot have more question or concenrs any more.

Reviewer #3 (Remarks to the Author):

The manuscript is well revised according to the reviewers' comments. It is recommended to be accepted.